# Hsp90 in Human Diseases: Molecular Mechanisms to Therapeutic Approaches

**DOI:** 10.3390/cells11060976

**Published:** 2022-03-12

**Authors:** Mamta P. Sumi, Arnab Ghosh

**Affiliations:** Department of Inflammation and Immunity, Lerner Research Institute, The Cleveland Clinic, Cleveland, OH 44195, USA; sumim@ccf.org

**Keywords:** angiogenesis, heme, heme-free, hemeprotein, oncoproteins, metastasis

## Abstract

The maturation of hemeprotein dictates that they incorporate heme and become active, but knowledge of this essential cellular process remains incomplete. Studies on chaperon Hsp90 has revealed that it drives functional heme maturation of inducible nitric oxide synthase (iNOS), soluble guanylate cyclase (sGC) hemoglobin (Hb) and myoglobin (Mb) along with other proteins including GAPDH, while globin heme maturations also need an active sGC. In all these cases, Hsp90 interacts with the heme-free or apo-protein and then drives the heme maturation by an ATP dependent process before dissociating from the heme-replete proteins, suggesting that it is a key player in such heme-insertion processes. As the studies on globin maturation also need an active sGC, it connects the globin maturation to the NO-sGC (Nitric oxide-sGC) signal pathway, thereby constituting a novel NO-sGC-Globin axis. Since many aggressive cancer cells make Hbβ/Mb to survive, the dependence of the globin maturation of cancer cells places the NO-sGC signal pathway in a new light for therapeutic intervention. Given the ATPase function of Hsp90 in heme-maturation of client hemeproteins, Hsp90 inhibitors often cause serious side effects and this can encourage the alternate use of sGC activators/stimulators in combination with specific Hsp90 inhibitors for better therapeutic intervention.

## 1. Introduction

In the year of 1962 the heat shock responses were first revealed by an Italian geneticist named Ferruccio Ritosa in the model organism Drosophilla [1]. Gradually it was determined that heat shock proteins (HSPs) are conserved and are ubiquitously found in the living world from bacteria to humans [2]. HSPs are multimolecular protein complexes expressed constitutively and comprise five to ten percent of the total cellular proteins under normal growth conditions. These proteins play multifunctional roles in cells, including intracellular transport in the cytosol, involving endoplasmic reticulum or the mitochondria, repair of degraded proteins, protein folding and refolding of misfolded proteins [3]. HSPs are distinctly induced by a range of environmental, pathological or physiological stimuli [4] and are controlled at a molecular level by factors such as oxidative stress, nutrient deprivation, hypoxia, apoptotic stimuli or neurocordal injury to the brain. According to their structure, function and molecular mass, HSPs are segregated into five major subgroups, i.e., HSP100, HSP90, HSP70 and the small HSP (sHSP)/α crystallins. The synthesis of HSP results from tolerance to insult, such as thermotolerance, or is induced by stress in various organisms [5]. In this review, we will discuss the diverse roles of the marquee chaperon HSP90 and its involvement in different human diseases.

Chaperon Hsp90 is constitutively expressed among eukaryotic cells which accounts for 1–2% of total cellular proteins, and these proteins can be significantly increased up to 4–6% under any physiological stress conditions [6]. The amino acid sequence of the Hsp90 protein is extremely phylogenetic, and is conserved from bacteria to humans [7]. In cells, Hsp90 associates with various substrate proteins, collectively called clients, and the chaperon is responsible for their proper maturation, activation and degradation. Essentially, most Hsp90 clients, including tau, synuclein, kinases, transcription factors, steroid hormone receptors, and E3 ubiquitin ligases are structurally or functionally diverse, where Hsp90 binds to these clients in different conformations [8]. Radli and Rüdiger give a current list of Hsp90 clients [9], and within this wide range of clients, inherent instability and low folding cooperativity seems to be the driving force for the chaperone activity of Hsp90. Along with Hsp90, there are a variety of cellular proteins which are the regulators of different intracellular cellular events, such as protein folding/degradation, cell growth, chromatin remodeling, cellular tracking, differentiation, etc., which may work in conjunction with the chaperon to enable such functions [6].

In mammalian cells there are two major isoforms of Hsp90, stress inducible Hsp90α and a constitutive Hsp90β [10]. Hsp90 homologs also exist in other cellular compartments, for instance Grp94 in the endoplasmic reticulum and Hsp75/TRAP1 into the mitochondrial matrix [10]. Structurally, Hsp90 contains three domains; the N-terminal ATP binding domain, the middle domain, and the C- terminal dimerization domain. The Hsp90 dimer undergoes conformational changes, distinct conformations being stabilized by interactions with co-chaperones and ATP. ATP hydrolysis drives the Hsp90 chaperoning cycle, and is essential for the proper folding of the proteins. Co-chaperones control the rate of ATP hydrolysis by Hsp90, stabilize certain conformational states or are involved in client recruitment [8,10]. The pleiotropic effects of Hsp90 on diverse client proteins means that Hsp90 is involved in many diseases, most prominently cancer, neurodegenerative diseases and some other respiratory diseases like PAH and asthma [11,12,13]. A number of HSP90 inhibitors have been identified as drugs that target the ATP-binding site or the C-terminal domain and a number of these are currently being assessed in clinical trials.

## 2. Structure and Functions of Hsp90

The molecular and structural features of Hsp90 have been reviewed analytically elsewhere [8,14]. Researchers revealed by their extensive studies that Hsp90 is present as a homodimer. According to the report, the monomer present in Hsp90 contains four domains including an N-terminal dimerization domain (NTD), a middle domain (MD), a charged region (CR) of a variable length, and a C-terminal domain (CTD) [14]. These four domains of HSP90 are amenably associated and the organization of the three domains of heterodimeric HSPs is preserved from bacteria to humans, but the CR domain is only present in eukaryotic HSPs. Stimulatingly, numerous preserved residues of the ATP-binding site in NTD preserve a “cap” which closes over the nucleotide-binding pocket in the ATP-bound state and is open in the ADP-bound state [15]. Studies described that these residues play a pivotal role in the ATPase activity of Hsp90 and are necessary for the chaperon cycle as well as for binding to client proteins. Thus, the CR domain is highly charged and comprised with a different length of amino acids which defines improved flexibility and dynamics to cope with the crowded environment in eukaryotic cells [16]. The middle domain (MD) of Hsp90 comprises critical catalytic residues to establish the compound ATPase site and interrelates with the γ-phosphate of ATP which stimulates ATP hydrolysis [17]. Furthermore, insights into client maturation have revealed that client proteins form contacts mainly with the MD of Hsp90, and also make contacts with the amino-terminal and carboxy-terminal domains. Moreover associated co-chaperones such as Aha1 binds with the N-terminal domain and to the MD to modulate the active conformation of the catalytic loop, which subsequently stimulates the ATPase activity of Hsp90 [10,17]. The CTD is responsible for the inherent dimerization and following the CTD is a highly conserved pentapeptide, MEEVD, which serves as the docking site for the interaction with co-chaperones containing a tetratricopeptide repeat clamp [18].

In cells, Hsp90 is involved in diverse cellular processes, including protein folding, post-translational modification, hemeprotein maturation, recognition, degradation, signaling transduction, cell cycle, and cellular differentiation [6,19,20,21]. Several studies have extensively reviewed the roles of Hsp90 in protein folding, maturation or prevention of protein aggregation and this field is ever-growing. Recently, the post-translational covalent modifications of Hsp90 have gained increasing significance. These modifications include phosphorylation, acetylation, nitrosylation, and methylation. Interestingly, these covalent modifications influence the chaperone activity of Hsp90 and, thus, the maturation of client proteins [22]. One widely accepted hypothesis on the recognition of Hsp90 for its clients is that Hsp90 recognizes certain conformations or the stability of selected clients rather than its primary sequence [23], and this may account for the diverse nature of Hsp90 function. However, further studies are needed to resolve this conundrum. Several reports have shown that Hsp90 is also required for facilitating protein degradation through involvement in the ubiquitin-proteasome pathway mediated by the carboxyl terminus of the Hsp70-interacting protein (CHIP) [24]. The clients of Hsp90 have been shown to associate with a large number of signaling proteins such as protein kinases and steroid hormone receptors [25]. Hence, Hsp90 seems to be essential in the maturation and protection of such protein function and is a key chaperone implicated in cell signaling. Hsp90 and its co-chaperones have been reported to be essential for the initiation and progression of diseases such as cancer, Alzheimer’s disease, asthma, pulmonary hypertension, neurodegenerative diseases, etc., [11,12,13,26]. Therefore, targeting Hsp90 in different ways can be a viable strategy to treat multidisciplinary diseases. Remarkably, the different phases of the cell cycle and differentiation intensely affects the mode of function of Hsp90 isoforms resulting in their different behavior based on the cellular ATP levels. As ATP binds to the NTD of Hsp90, the hydrolysis of ATP provides energy for its functions. Therefore, competitively inhibiting the ATP-binding process with Hsp90 inhibitors has gained significant attention in disease treatment, and several Hsp90 inhibitors have recently entered clinical trials [27].

## 3. Conformation Dynamics of Hsp90 

Extensive structural studies have revealed that the Hsp90 chaperone cycle could be divided into distinct conformations, which seem to be in a dynamic equilibrium [28]. In the apo state (Figure 1), Hsp90 adopts an open V-shaped form predominantly, termed “open conformation” [18]. Then, the binding and hydrolysis of ATP drives the conformational changes and leads to the formation of the first intermediate state, in which the ATP lid is closed, but the NTD is still open [29]. The unfolded substrates are recognized by co-chaperones and loaded onto Hsp90 during this process [30]. Subsequently, Hsp90 ATPase activity is triggered by co-chaperone Aha1 along with the dissociation of other co-chaperones, such as Hsp70 and Hop, which promotes the closure of the Hsp90 homodimer, and this brings Hsp90 in its “closed conformation” [31]. This dimerization leads to the formation of the second intermediate state, in which the MD repositions and interacts with the NTD. The ATP hydrolysis is catalyzed in this fully closed state, resulting in substrate folding. Then, the subsequent opening of the NTD of the Hsp90 dimer restores the initial “open conformation” [29,32].

## 4. Role of Hsp90 in Client Hemeprotein Maturation

Hemeproteins are ubiquitously distributed metalloproteins found in nature which contain heme or an iron (Fe) protoporphyin IX (or a derivative of this macrocycle) in the active site [33]. Based on the side chains present in the macrocycle and the type of bond involved in the interaction with the protein, the hemes are classified in three major classes; a, b and c [34]. Hemes a and c are synthesized from heme b, which is the most abundant heme found in hemeproteins, and distinctive chemical modifications occur in the side chains (Figure 2A). The heme is coordinated to the protein by axial ligation through histidine, methionine, tyrosine or cysteine residues [34]. The distinctive classes of heme proteins (Figure 2B) play pivotal roles in different biological functions including cell energetics (cytochromes), oxygen transport/storage (hemoglobin and myoglobin), hydrogen peroxide degradation (peroxidases, catalases), oxygen reduction (heme-copper oxidases), heme or small molecule sensing (FixL, HemeAT sensors), transcription regulation (Per2, RevErb αβ, Bach1), cell signaling (sGC, NOSs), numerous enzymatic transformations, etc., [35,36,37] and remarkably use the same heme cofactor to confer such diverse functions. Thus in all cases, the heme cofactor is essential for function, but because free heme is toxic its production is tightly regulated [38].

The specific steps of heme biosynthesis are well-documented and take place in both the cytosolic and mitochondrial compartments of a cell, [39,40,41] with the final three steps occurring in the mitochondria. However, with the exception of cytochrome c biogenesis [42] and some aspects of heme acquisition and catabolism [43,44] relatively little is known about how heme is transported out of mitochondria in eukaryotic cells and how it becomes inserted into soluble proteins in the cytosol, or how these processes might be regulated [45]. These steps are critical, given that free heme is potentially cytotoxic [46] and is normally kept at low intracellular levels [47]. New studies have identified a mitochondrial heme exporter FLVCR1b, which is essential for erythropoiesis [48,49]. How this heme exporter pertains to heme transport into soluble hemeproteins in the cytosol remains to be investigated. Studies from our group [50,51] as well as some from Osawa’s group [52] have revealed a specific involvement of chaperon Hsp90 in cytosolic heme insertion into soluble proteins such as iNOS and soluble guanylate cyclase (sGC) (Figure 3). In both cases we found that Hsp90 associates with the heme-free, apo-enzyme (heme-free or apo-iNOS monomer or apo-sGCβ1 subunit) in cells, and then drives heme insertion into the apo-enzyme by an ATP-dependent process, after which the Hsp90–apo–enzyme interaction falls apart. Since our models for Hsp90 function in sGC and iNOS maturation are identical [50,51], it implies that Hsp90 may operate through a common mechanism to target and stabilize the heme-free forms of client hemeproteins, and then enable their maturation by driving heme insertion by an ATP-dependent process (Figure 3). Our past [20] and present [53] findings also revealed that low doses of nitric oxide (NO) can contribute to sGC maturation by triggering a rapid Hsp90-dependent heme insertion into the apo-sGCβ1 (heme-free) population, ultimately resulting in a mature sGC-α1β1 heterodimer [20]. This finding of an elevated active sGC-α1β1 heterodimer formation by a NO trigger causing heme-insertion into apo-sGCβ1, filled a void in an earlier work done by Ignarro and colleagues in the 1980s which showed that NO-heme moiety could be transferred into heme-free or apo-sGCβ1 through an exchange reaction with NO-hemeproteins to activate the enzyme [54]. This suggests that a NO-Hsp90 synergy may be essential for maturation and activation of certain hemeproteins. Other studies on transmembrane heme proteins such as NADPH oxidases (NOX) have shown preliminary evidence of a Hsp90 [55]. NOX enzymes utilize NADPH to synthesize superoxide. This NOX functionality is dependent on its heme and is a major source of cellular reactive oxygen species (ROS) [56,57]. Due to the intrinsic toxicity of ROS, the regulation of NOX enzymes has developed a significant degree of complexity. Hsp90 has been shown to bind to the C-terminal domain of NOX5 and regulate its superoxide production, suggesting that Hsp90 may regulate certain parameters of cellular redox [55]. More recently, a study identified a more dynamic regulatory control over NOX5 activity by modulating its heme via intracellular heme levels and Hsp90 [58], however, such studies require physiological semblances and follow up studies are needed. 

New research from our group has also identified Hsp90 and GAPDH to be major players in the process by which hemoglobin (Hb) and myoglobin (Mb) form and mature [21,59,60]. We found that Hsp90 chaperones hemoglobin (Hb) maturation in erythroid and non-erythroid cells (RAW, A549 cells) following a similar mechanism [21] (Figure 4). While in erythroid cells, Hb-α and Hb-β/γ (-β_adult_/γ_fetal_ Hb) are independently chaperoned by AHSP (alpha Hb stabilizing protein) and Hsp90, respectively, and Hsp90 promotes heme-maturation of both Hb-αβ subunits in non-erythroid cells which lack AHSP. We also found that muscle Myoglobin (Mb)/Hb not only requires Hsp90 and its co-chaperon machinery, but also needs an active sGC for its heme-maturation [59] (Figure 4). We recently established that in all these globin maturation events, GAPDH acts as a heme chaperon, allocating/providing mitochondria generated heme to the apo-Hsp90/AHSP-Hb**_α/β/γ_**/Mb complexes [60], where these globin heme maturations are influenced by iron provision, magnitude of expression of GAPDH, d-aminolevulinic acid plus FLVCR1b and that Hsp90 may be the ultimate heme donor to these globins [60]. Together these finding imply that Hsp90 and GAPDH play key roles in globin maturation and our current studies link the activation of sGC to the maturation of the globins (Mb/Hb), thereby contributing to the formation of novel NO-sGC-Globin axes. Our studies may find potential use in developing treatments for cancer, since some cancer cells utilize Hb differently than red blood cells [61,62] and other blood diseases such as sickle cell anemia, thalassemia, etc., [63,64,65].

## 5. Hsp90 and Human Diseases

As a ubiquitous chaperone, Hsp90 has received much attention due to its significant roles in regulating proliferation, growth, differentiation, adhesion, cancer metastasis, angiogenesis, and apoptosis. Thus, Hsp90 is perhaps one of the most widely studied heat shock proteins [66] as it is at the epicenter of human diseases, including neurodegenerative diseases, pulmonary/respiratory diseases (pulmonary arterial hypertension, acute pulmonary fibrosis, asthma) and different types of cancer [12,13,67,68,69,70]. The ability of the chaperon to protect unfolded proteins from aggregation, assist in proteosomal degradation and modulate several growth and signal pathways simultaneously, makes it an attractive therapeutic target [71]. Higher levels of chaperon Hsp90 has been found in a wide spectrum of cancers, suggesting a central role in the survival and growth of malignant cells [72,73,74]. Such enhanced levels of Hsp90 is attributed to having a protective effect (via regulation of HSF-1) from various stress parameters such as hypoxia, ischemia, etc., which the cells encounter under pathologic conditions, and since several oncoproteins are clients of Hsp90, targeting Hsp90 represents a useful anti-cancer approach.

## 6. Role of Hsp90 in Cancer/Tumorogenesis

Hsp90 and its chaperon proteins are involved in multiple cellular signaling pathways which regulate apoptosis and promote cell survival [68]. In this context, three acquired capabilities of cancer cells’ growth and sustenance are evasion of apoptosis, sustained angiogenesis and tissue invasion [71]. A great body of the literature indicates a critical inhibitory contribution of Hsp90 to apoptosis which is key to normal cell growth or adverse tumor progression. Hsp90 is known to inhibit apoptosis by directly binding to Apaf-1, blocking cytochrome c-mediated oligomerization of Apaf-1 and the activation of pro-caspase 9 [75]. It also inhibits apoptosis by forming a ternary complex with pro-apoptotic kinase Ask-1 and Akt [76]. Moreover, Hsp90 exerts anti-apoptotic activity by blocking the mitochondrial-cytosolic transition of the apoptosis-inducing factor (AIF) and endonuclease G [77]. With regard to angiogenesis, Hsp90 is known to promote angiogenesis and metastasis by chaperoning certain client proteins including VEGF, NOS, and MMP-2 [73]. Together these functions of chaperon Hsp90 make it clearly anti-apoptotic and pro-angiogenic [71].

Various forms of cancer propagate due to an elevated Hsp90 expression, elevated Hsp90 ATPase, significant association of Hsp90 in the regulation of different client/oncoproteins, and due to its anti-apoptotic properties. Examples of such manifestations are diverse including (but are not limited to) colorectal, prostate, breast and lung cancer. ***Colorectal cancer*** is one of the most common malignancy types worldwide with a poor survival rate [78]. Owing to the function of Hsp90 in the regulation of different oncoproteins, it has become a striking therapeutic target for different cancer types including colon cancer [72]. In ***Prostate cancer*** the androgen receptor (AR) plays a significant role in the development, maintenance and progression of prostate cancer [79]. Here also, oncoproteins, such as the AR, depend on Hsp90 to maintain their cellular functions and rapid differentiation [80,81]. In prostate cancer, Hsp90 inhibitors promote the degradation of AR to inhibit the oncogenic activity of AR [80]. Hence, targeting Hsp90 could be a beneficial therapeutic strategy to treat androgen-dependent PC. ***Metastatic Breast cancer*** signifies the second most common cause of female mortality and morbidity [82]. Here, the chaperon interacts with different proto-oncogenes essential for the growth and survival of such cancer cells, eg., progesterone receptor (PR), tumor suppressor proteins P5, estrogen receptor (ER), angiogenesis transcription factor HIF-l alpha, anti-apoptopic kinase AKT, RAF-1 MAP kinase and variety of kinases of ErbB family including Her2/neu, as well as proteins downstream of Her2/neu [71,83]. Hsp90 also protects the mutation of amino acid from proteasomal degradation, for instance, the mutation of the V599E amino acid sequence in B-RAF protein is protected by Hsp90, but the wild type is not [84]. In ***Lung cancer*** an increased expression of Hsp90 is found in NSCLC (Non-small cell lung cancer) patients and increased Hsp90 positively correlates with the age, lung squamous cell carcinoma (LSCC), ever-smoking history and metastasis of lymph node. Furthermore, a raised Hsp90 expression found in NSCLC patients which is significantly associated with smaller survival, suggests that it could be used to predict survival [85]. In addition, a high expression of Hsp90 is significantly associated with male patients and those with a smoking index over 600 in SCLC [86]. Moreover, a study reported that the overall survival rate of high Hsp90β expression in lung cancer is shorter than that of the low expression Hsp90 group and that Hsp90β can be an independent prognostic factor [87]. Moreover, an elevated Hsp90β was found in the serum and was highly associated with the differentiated grade and advanced clinical stage of lung cancer patients and contributes to a novel diagnostic and prognostic biomarker [88]. These studies suggest that Hsp90β can be an independent prognostic biomarker and that the high expression of Hsp90 in these cancer types maybe inversely related to survival rates. 

The tumor selectivity of Hsp90 inhibitors makes Hsp90 a unique therapeutic target. What stands out here is the high ATPase of tumor Hsp90 which may be the driving force as we see in our heme-maturation of client hemeproteins under normal conditions [50,51]. The ability of cancer cells within tumors to expand requires angiogenic ability and neovascularization. Since such angiogenic pathways are turned on during tumor progression [89], the vascularization machinery becomes active enough to sprout new blood vessels to sustain neoplastic growth. As new blood vessel growth requires functional Hsp90 [71], active heme-maturation of Hsp90 clients, such as iNOS [50], Hb and sGC [21,51], may also occur during angiogenesis within these malignant tumors to support these processes. These events can now be better envisioned based on our signature finds.

***Implications of an active NO-sGC-Globin axis in cancer cell metastasis*****:** In general, both Hb and Mb expression in tumors has been attributed to counter/scavenge effects of hypoxia or oxidative stress prevalent in these tumors [61,90]. There are two possible roles of NO/cGMP signaling in malignant tumors. Its known that iNOS-expression and NO overproduction may contribute to the formation of an inflammatory cancer micro-environment which derives angiogenesis, as in the case of gliomas [91,92]. Second, sGC/cGMP signaling may influence proliferation and/or differentiation of the tumor cells. Since our studies link globin expression to the NO-sGC axis [21,59,60], it provides additional ways to eliminate such globin expression. Inhibiting iNOS overexpression and the tumor inflammatory microenvironment, together with normalizing sGC/cGMP signaling, may be one of the favorable ways to reduce malignant tumors [91,92]. Cancer metastasis is the leading cause of cancer-related death, and Circulating tumor cells (CTCs) are cancerous cells that are shed from either primary or metastatic tumors during an intermediate stage of metastasis into the bloodstream. CTCs flow across the bloodstream avoiding the immune system by a process called epithelial to mesenchymal transition (EMT) so that they do not get consumed by phagocytes and then can metastasize into another part of the body by extraversion (leaving the bloodstream) [93]. While it is becoming increasingly clear that CTCs are heterogeneous at multiple levels and that only a fraction of them are capable of initiating metastasis such that the tumors spread to other parts of the body [93], it also appears that CTCs adopt multiple ways to enhance their metastatic potential [93]. While some studies state the role of a lung derived Hbβ to be anti-metastatic for neuroblastoma cells [62], more studies in CTCs suggest evidence which is contrary [61,94]. For example, it was found that asymmetric Hbβ expression in a variety of tumor cells and in the CTCs increased their metastatic potential, possibly by enhancing tumor-cell survival during blood-born dissemination [61]. This discovery highlights the importance of the NO-sGC-Globin axis where a globin is needed for cell survival in aggressive cancer/tumor cells. Interestingly, our studies revealed that Hsp90 is needed for the heme-maturation of Hbβ and not Hbα [21], which would increase the selectivity of the Hsp90 inhibitors to block Hbβ heme-maturation [21]. Moreover, since there are multiple pathways of regulation to maintain the NO-sGC-Globin axes, it offers encouraging check points where this pathway could be stalled by therapeutic intervention to curtail the expression of the globins which may in turn inhibit the metastatic potential of these tumors. Thus, inhibiting the NO-sGC signaling in these CTCs, and thus restrain Hbβ maturation, can be a favorable way to stop the spread of malignant tumors. 

## 7. Role of HSP90 in Neurodegenerative Disorders

***Alzhimers disease (AD)*** is defined as the development of the extracellular accumulation of amloid-β (Aβ) in senile plaques and intracellular neurofibrillary tangles (NFTs). The protein fragment Aβ is produced from an amyloid precursor protein (APP) and stored as insoluble plaques in solid form. The NFTs are made up of microtubule associated protein, tau. Hence, microtubules play a significant role in maintaining the neuronal morphology in AD tau protein becoming abnormal and microtubule structural collapse [95,96]. In AD, hyperphosphorylation of tau protein occurs resulting in an elevated tau protein accumulation and marks microtubule destabilization, which in turn then leads to neurodegeneration [97,98]. As tau is a client protein for the Hsp90 chaperon complex, when it is abnormal or modified, it induces the CHIP protein in the Hsp90 complex and encourages the ubiquitination of the tau protein, leading to its degradation [99] (Figure 5). Another pathological characteristic of AD is the fibrillary Aβ peptide accumulation in the nerve cells of the brain leading to amyloid plaque formation. The amyloid-β formation can be inhibited by Hsp90 and co-chaperones that decrease the accumulation in a chaperone concentration dependent manner [100]. Studies have reported a different mechanistic inhibition of Aβ protein aggregation by Hsp90 ATPase activity, such as either Hsp90 binding with a misfolded amyloid protein and prevented from aggregation or modifying the conformation of the protein by binding with Aβ, resulting in a reduced vulnerability to aggregation [100,101]. ***Parkinson’s disease (PD)*** is the next common neurodegenerative disease after Alzheimer’s disease, categorized by postural instability, stiffness, bradykinesia, active tremors, etc. [102,103]. Studies have identified different proteins associated to PDs including alpha-synuclein (α-syn), Parkin, PTEN induced putative kinase 1 (PINK1), leucine-rich repeat kinase 2 (LRRK2), etc., [104]. α-syn is modulated by HSP90 in an ATP dependent manner, Hsp90 stimulates the growth of protein fibrils but it stabilizes the toxic pre-fibrillary oligopeptide in the absence of ATP [105]. The HSP organizing protein (HOP), and other co-chaperone of Hsp90 mediate the transfer of clients to HSP90 and inhibit the ATPase activity of Hsp90, which might promote the formation of soluble oligomer [106]. Some researchers suggested that, when Hsp90 is inhibited, the Hsp90 complex releases a heat shock transcription factor (HSF-1) and increases Hsp70 expression leading to a Hsp70 dependent ubiquitination process [107]. Moreover, the ubiquitination directly depends on the interaction between Hsp90 and its substrates, for instance, the client proteins of Hsp90 are swiftly degraded by the ubiquitinin proteasome system (UPS) pathway if the Hsp90 complex is blocked. This concept might unveil a therapeutic strategy for cancer. An in-vitro study conducted by Falsone and his team in 2009, discovered that Hsp90 inhibition or blockade is directly involved in the blocking of α-syn aggregation. In Parkinson disease (PD), the reduction of oligomeric α-syn occurs due to the inhibition of Hsp90 resulting in a lesser cytotoxicity of the mutant A53 α-syn, which leads to increased dopamine [102,108], specifying the Hsp90′s direct interaction with soluble α-syn protein, and the inhibitors of HSP90 stimulate soluble client degradation by UPS before the proteins aggregate (Figure 5). Therefore, inhibitors of Hsp90 could be potential therapeutic targets. Although, there are different Hsp90 inhibitors being evaluated in clinical trials against cancer [109], but the application of the inhibitors in neurodegenerative diseases are yet to be discovered.

## 8. Role of Hsp90 in Asthma and Pulmonary Diseases

The ability of Hsp90 to modulate cell fate might have vital repercussions not only for cancer, but also for other progressive human diseases, such as those of the pulmonary system, eg., ***Asthma*** [110] and ***Pulmonary fibrosis*** (PF) [111]. ***Asthma*** is defined by airway inflammation and hyper responsiveness, and contributes to morbidity and mortality worldwide [110]. sGC is a key enzyme of the NO signaling pathway and is activated by NO produced from NOS enzymes [50,112], thereby activating synthesis of the second messenger cGMP, which produces vascular smooth muscle relaxation or vasodilation as a downstream effect [113,114]. This constitutes the NO-sGC-cGMP signal pathway and is a well-known dilation pathway in the vasculature. In a study [12], we found that the NO-sGC-cGMP pathway plays a significant role in lung bronchodilation, and that lung sGC becomes dysfunctional in asthma due to high levels of NO generated from iNOS induction during inflammation in the airway epithelium [12,115] which desensitizes the sGC resident in the airway smooth muscles below. This dysfunctional sGC is heme-free, does not respond to its natural activator NO, but can be activated by sGC agonists such as BAY 60-2770 [116], which can activate the enzyme independent of NO to produce bronchodilation, and such agonists are future drugs for asthma. In this study, Hsp90 was associated with heme-free sGC, and seemed to stabilize it rather than priming it for degradation, which makes this pathological sGC more drug receptive. In other studies on asthma relating to airway epithelium and inflammation, Hsp90 is more directly implicated. Extracellular Hsp90α (eHSp90α) is shown to mediate HDM-induced human bronchial epithelial dysfunction, suggesting that eHsp90α is a potential therapeutic target for the treatment of asthma [117]. Another report [118] suggests that Hsp90 inhibition reverts IL13 and IL-17-induced goblet cell metaplasia in human airway epithelia. Hsp90 inhibitors have also been shown to reduce airway inflammation in mouse models of allergic asthma and another study showed that Hsp90 inhibitors hampered airway relaxation [119]. Given our current understanding of iNOS heme-maturation, most of these processes of Hsp90 inhibition may involve inhibition of iNOS heme-insertion, suppressing iNOS induction in the airway epithelium and reducing inflammation [12].

Lately, a number of reports have depicted the involvement of Hsp90 in the pathogenesis of ***Pulmonary arterial hypertension*** (PAH), particularly in vascular remodeling, despite that the defined pathogenesis mechanisms of PAH have yet to be elucidated [13,120]. Many studies described that an increased level of Hsp90 was observed in both the plasma and membrane walls of pulmonary arterioles in PAH patients [13]. Furthermore, a small molecule 17-AAG, the inhibitor of Hsp90, has been used to inhibit pulmonary vascular remodeling as this may suppress the proliferation and migration of Human pulmonary artery smooth muscle cells (PASMCs). Other findings indicate that the accumulation of Hsp90 in PASMC mitochondria was a hallmark of PAH development and a basic regulator of mitochondrial homeostasis which contributes to vascular remodeling in PAH. Not astoundingly, cytosolic Hsp90 stimulates PASMC proliferation by stabilizing key signaling proteins involved in PAH development and progression. Additionally, different study results demonstrate that environmental stresses promote vascular remodeling and mitochondrial accumulation of Hsp90 in PASMCs of PAH patients which aid in proliferation and survival [120]. Gamitrinib, which is a small molecule designed to target selectively Hsp90 in mitochondria, was related with antiproliferation activity in preclinical models with no obvious organ or systemic toxicity [121]. Furthermore, it was established that the targeted inhibition of mitochondrial Hsp90 with Gamitrinib reversed pulmonary vascular remodeling and improved cardiac output in two PAH models without perceptible toxicity. Hence, pharmacological inhibition of Hsp90 is a propitious possibility to improve the clinical outcomes of patients with PAH, and drugs targeting Hsp90 in mitochondria may further show advantages in PAH treatment [120].

***Idiopathic pulmonary fibrosis (IPF)*** is a progressive disease of the lung parenchyma, causing significant morbidity and mortality [122,123]. Transforming growth factor (TGF)-β1 is a key cytokine involved in the process of fibrogenesis [124]. It causes myofibroblast proliferation and differentiation and increases the synthesis of collagen, fibronection and other extracellular matrix components (ECM). A recent study [125] demonstrated that Hsp90 has a direct role in the TGF-β1 signaling pathway and Hsp90 inhibition reduced lung fibrogenesis and fibrosis progression in mice. The study also showed that Hsp90 is overexpressed in IPF lungs and fibrosis can be inhibited using a water soluble Hsp90 inhibitor, 17-DMAG, which targets the Hsp90 ATPase similar to 17-AAG [125]. Moreover, studies with second generation water soluble Hsp90 inhibitors, such as AUY-922, have shown that it effectively blocks fibrosis in murine models of nitrogen mustard (NM)-induced pulmonary fibrosis [126]. Here, AUY-922 suppressed NM-induced inflammation in mice lungs, reduced the expression and accumulation of extra-cellular matrix proteins, removed histological evidence of fibrosis and restored normal lung function [126]. Studies with AUY-922 also showed that it is effective against lung injury caused by hydrochloric acid or radiation [127,128], both of which can cause acute inflammation leading to fibrosis. These studies display that AUY-922 blocks the adverse effects on lungs from such exposures and represents a promising approach against the development of pulmonary fibrosis. Other studies by Sontake et al. [129] found that Hsp90 was elevated in expression and in its ATPase activity in lung biopsies of patients with IPF. These findings bear great semblance with regard to Hsp90 expression and ATPase to what we see in a majority of cancers [71] and suggests that targeting Hsp90 is an effective strategy for treating fibrotic lung disease. More recently sGC agonists [130] are also being tried to treat age-related fibrosis. Since Hsp90 and other chaperon machinery slows down with age [131], causing low NO levels which may reduce sGC activation, thereby making such sGC agonists as the drugs of choice. Moreover, a slow Hsp90 chaperon machinery may cause reduced heme-maturation causing a buildup of heme-free sGC, which can be activated by BAY 60-2770 like drugs (Figure 6), further encouraging the use of these agonists to treat age-related fibrosis.

## 9. Therapeutics Facets of HSP90 Inhibition 

Hsp90 has emerged as an important molecule in anti-tumor therapy, and several drug classes have been found to target its N-terminal ATP-binding domain resulting in inactivation of the chaperone. The classic targeting of its N-terminal domain (NTD) began with natural analogs such as geldanamycin, herbimycin, etc., [132]. Later, a semi-synthetic derivative of geldanamycin (GA), called tanespimycin (17-AAG), was made which improved the ADME activity and reduced the toxicity effects relative to GA. In 1999, 17-AAG entered clinical trials, while a second inhibitor, 17-DMAG, entered the first in-human study in 2004 [68]. From this period onwards, many other Hsp90 inhibitors have been and are currently undergoing clinical evaluation in cancer patients, owing to extensive efforts in rational drug design and discovery [133,134]. Moreover, various other newer classes of Hsp90 inhibitors are also on the rise [109,135]. While the inhibition of Hsp90 via the NTD perturbs the repressive effects of Hsp90 on HSF-1, which then translocates into the nucleus to activate a heat shock response by inducing the transcription of Hsp70, Hsp40 and Hsp27 genes which have powerful anti-apoptosis, drug resistance, and proliferative effects [136]. Hence the upregulation of this heat shock response allows cells to develop resistance and dampen the effect of potent Hsp90 inhibitors. An alternate driving force is the strategy to lessen this heat shock response by use of the C-terminal domain (CTD) targeting Hsp90 inhibitors. This class of inhibitors fall into two categories: inhibitors that directly target the C-terminus and inhibitors that disrupt the binding of Hsp90 to co-chaperons that bind to the CTD of Hsp90 [137]. Like the NTD, the CTD also contains the ATP binding site but lacks ATPase activity [8]. Selective targeting of this binding site has shown promising applications in inhibiting chaperon function [138,139] and Coumarin-based antibiotics were among the first inhibitors to target the CTD of Hsp90 [139]. Inhibitors such as Novobiocin, a coumarin-antibiotic, have been shown to affect the stability of Hsp90-client protein and release protein interactions by disrupting the dimerization of Hsp90 [139,140]. Since then, various Novobiocin analogues have been developed which show strong anti-proliferation activity in prostate cancer lines [139,141] or show potential neuroprotective properties and provide avenues where Hsp90 inhibition can be studied in the context of neurodegeneration [138]. Since Hsp90 was identified as a potential cancer target, various inhibitors targeting cytoplasmic Hsp90 have been established as anticancer agents [73,142,143], and many different inhibitors with desirable pharmacological properties, including ganetespib (STA-9090), have been evaluated in clinical trials, but none have shown satisfactory efficacy to be approved by the FDA. The main reason for this limited efficacy can be attributed to the dose-limiting toxicity of the drug and the inevitable activation of HSF-1, leading to cytoprotective heat shock responses by the upregulation of prosurvival genes [8,144]. In order to circumvent these shortcomings, various newer strategies are coming up including the coinhibition of HSF-1 along with Hsp90 [145], and this may hold great promise in future therapeutics. Table 1 summarizes the various Hsp90 inhibitors used following different mechanistic approaches to target Hsp90 and/its co-chaperons.

A wealth of evidence now indicates that the successive dosing of Hsp90 inhibitors to animals bearing human tumors blocks tumor growth efficiently. However, tumors often re-grow after the withdrawal of the inhibitor [146]. A similar observation is made in patients with solid tumors treated with several structurally unrelated Hsp90 inhibitors, thus rendering Hsp90 inhibitors an unsuitable agent in monotherapy. However, in certain tumor entities, single-agent inhibitor therapy was found to suppress tumor growth [147]. Likewise, Hsp90 inhibitors in combination with cutting-edge targeted therapies is the pathway to move forward. It has been shown that a combination of the Hsp90 inhibitor SNX-5422 and trastuzumab (herceptin), a monoclonal antibody that blocks the Her-2 receptor, led to a synergistic regression of tumor growth in a xenograft model of human breast cancer [146]. As another example, a phase II trial, in which the combinatorial administration of 17-AAG (tanespimycin) plus trastuzumab displayed significant anti-cancer activity in patients with Her-2-positive metastatic breast cancer, which was previously seen to progress on trastuzumab [148]. Thus the synergistic effects in tumor regression observed in animal studies after the combinatorial administration of Hsp90 inhibitors and potent anti-cancer drugs hold true in human trials, hence targeted combination therapies hold potential for future drug designing. Based on our current knowledge of hemeprotein maturation and the role played by Hsp90, the use of Hsp90 inhibitors which target the ATPase function of the chaperon can also give rise to certain side effects (Figure 6). These inhibitors can adversely affect normal cells by blocking the heme-maturation of iNOS, sGC or Hb (Figure 6). For example, the Hsp90 inhibitors that are being developed for cancer treatment might unintentionally block Hb maturation in the recipient. Indeed, anemia has been commonly reported as a side effect during the clinical trials of Hsp90 inhibitor drug candidates [149,150]. Likewise, the Hsp90 inhibitors being used for asthma, IPF or PAH may also block heme-maturation of sGC, creating pathologic heme-free sGC, and consequently reduce sGC activation in these diseased conditions. This can in turn worsen these conditions and obstruct bronchodilation cascades. Hence, widespread use of these Hsp90 inhibitors may require a more cautious approach or a combination therapy with sGC agonist like drugs may help in future therapeutics in such cases (Figure 6). 

***Hsp90 as a therapeutic target in zoonotic diseases:*** Hsp90 has been shown to play critical roles in the life cycle of various pathogenic protozoans and has been shown to regulate cellular processes in zoonotic protozoan parasites such as *Plasmodium, Giardia, Toxoplasma, Trypanosoma, Leishmania*, etc., [151,152,153,154]. As these pathogens cause several infectious diseases in humans, Hsp90 is a key drug target in such infectious diseases [155,156]. Most notably, the geldanamycin water soluble analog 17-DMAG has been shown to be an effective anti-trypanosomal drug on in-vitro cultures and mouse models of *Trypanosoma brucei* [157]. It was also reported that 17-DMAG inhibits the multiplication of several *Babesia* species and *Theileria equi* [157]. For diseases spread by viruses, a study found that the phosphoprotein (P) of the rabies virus (RABV) is chaperoned by the Cdc37/Hsp90 machinery during infection [158], and this may have significant potential for developing Hsp90 inhibitors against Rabies. Additionally, Ron Geller and his team reported that Hsp90 inhibitors may present attractive antiviral therapeutics for the treatment of RSV (Respiratory syncytial virus) infections [159] as Hsp90 inhibitors degrade the viral protein during infection, and this shows the potential of chaperone inhibitors as antivirals exhibiting high barriers to the development of drug resistance during such viral infections. More recently, transcriptomic profiling studies on SARS-CoV-2 infected human cell lines identified Hsp90 as a target for COVID-19 therapy [160]. While many of these Hsp90 inhibitor drugs may display host cell toxicity, further combinations with other drugs can improve the outcome while simultaneously reducing toxicity.

## 10. Conclusions

There are at least 18 Hsp90 inhibitors [145] currently undergoing clinical trials from a broad range of tumors and in vitro studies continue to identify newer small molecule compounds that selectively target the Hsp90 chaperon in its various paralog forms [161,162]. Moreover, the chances of drug resistance decrease with the application of rationale drug combinations, for the inhibition of Hsp90 offers a wide range of opportunities to enhance the anticancer effects of drugs used in combination therapies. Thus, this ever-growing progress of Hsp90 inhibitor drugs and their relevance to specific diseases holds great promise for further exciting developments in the future. Given the novel role of Hsp90 in hemeprotein maturation, the effects of an overactive or a downregulated Hsp90 can both be deleterious to cellular homeostasis. This is now more evident from the fact that Hsp90 regulates the heme maturation of three key hemeproteins (iNOS, sGC and Hb) and our studies [21,59,60] now link the activation of sGC to the maturation of the globins (Mb/Hb), thereby contributing to the formation of novel NO-sGC-Globin axes and brings this pathway into a “new light” whose significance has thus far not been explored in cancer cells. Since Hsp90 is central to the heme maturation of iNOS, sGC and globin, its well defined role in cancer progression needs to be critically looked at cautiously with a fresh perspective so as to overcome potential deleterious effects of various Hsp90 inhibitors for better therapeutic intervention. Together, these realizations provide a platform to explore these concepts which may help in future drug designing. 

## Figures and Tables

**Figure 1 cells-11-00976-f001:**
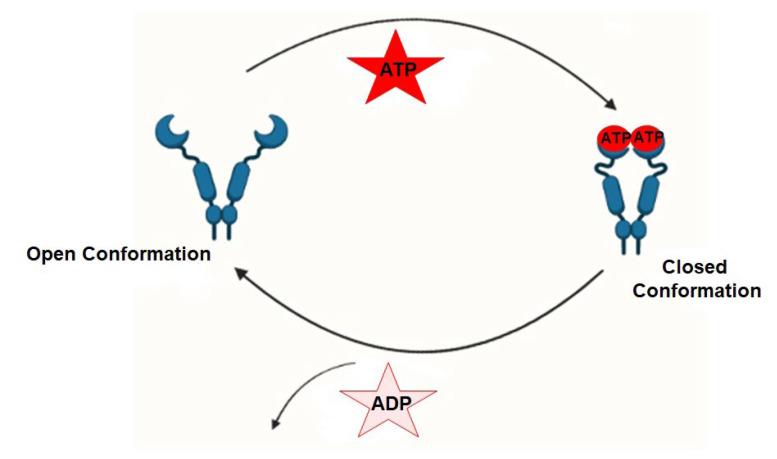
Representations depicting the conformational dynamics of Hsp90 (open and closed states) brought about by its ATPase function during substrate/client protein folding.

**Figure 2 cells-11-00976-f002:**
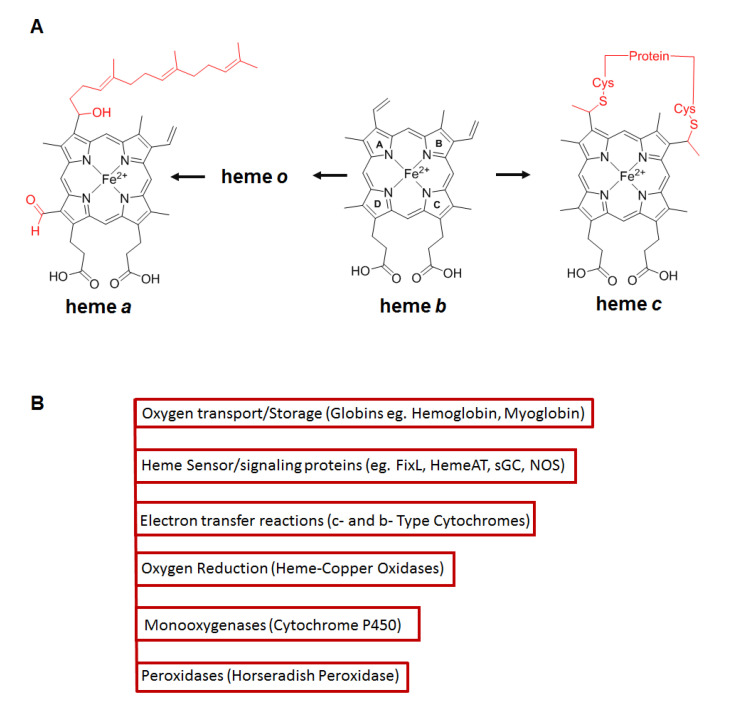
(**A**) Structures for heme a, b and c. Heme a and c are synthesized from heme b via side chain modifications shown in red. Pyrrole rings nomenclature a, b, c and o are depicted using the Hans Fischer system. (**A**) Adapted from Reference [33]. (**B**) Representative classes of heme proteins which incorporate such hemes.

**Figure 3 cells-11-00976-f003:**
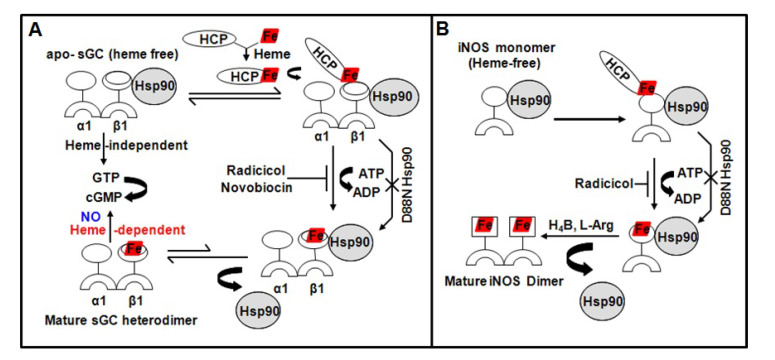
Representations depicting similarities in Hsp90–apo–protein interactions with respect to heme deplete/replete states of sGC (**A**) and iNOS (**B**). HCP indicates an ultimate heme carrier protein which may eventually be Hsp90.

**Figure 4 cells-11-00976-f004:**
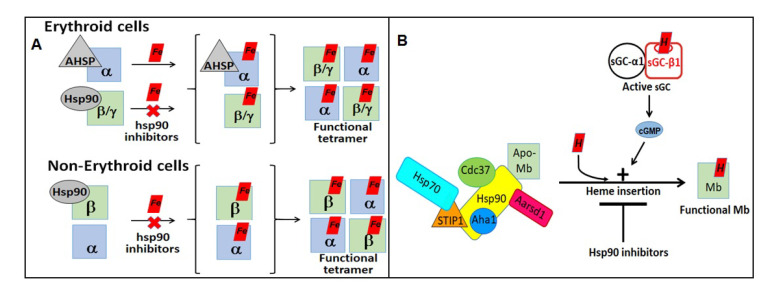
(**A**,**B**) Representations depicting the role of Hsp90 in hemoglobin (Hb) and myoglobin (Mb) maturations.

**Figure 5 cells-11-00976-f005:**
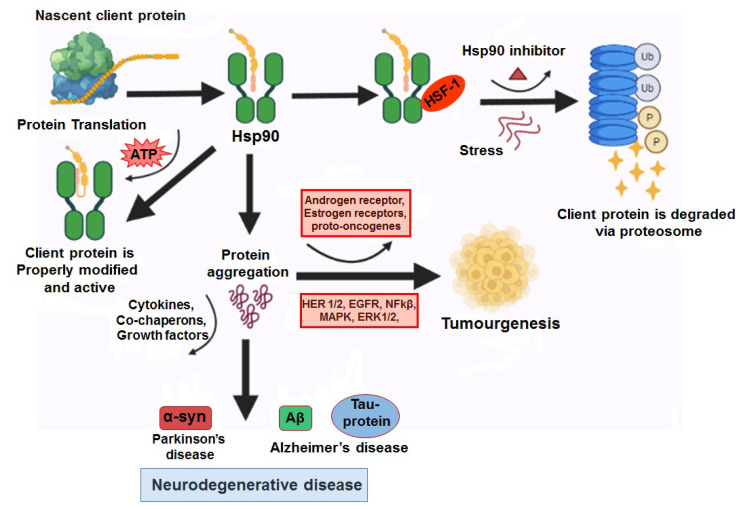
Representations depicting the role of Hsp90 and associated co-chaperons in cancer and neurodegenerative diseases.

**Figure 6 cells-11-00976-f006:**
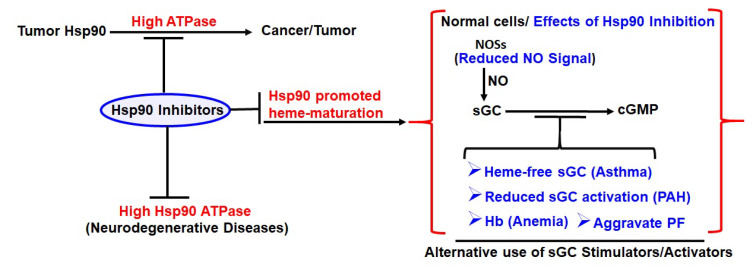
Representations depicting the potent effect of Hsp90 inhibitors on the high Hsp90 ATPase in cancer or neurodegenerative diseases and the adverse effects of these inhibitors in normal cells due to inhibition of heme-maturation of depicted proteins.

**Table 1 cells-11-00976-t001:** List of various Hsp90 inhibitors used in different diseases which impact distinctive mechanistic pathways.

Disease	Hsp90 Inhibitors	Mechanism	Results
Colorectal cancer	Cetuximab	VEGF/VEGFR signaling pathway	Block angiogenesis
	Panitumumab		
	Bevacizumab		
	Regorafenib		
	Ziv-aflibercept		
Prostate cancer	Ganetespid	PI3K/mTOR signaling pathway	Tumour cell death Growth inhibition
Brest cancer	17 AAG (Demethoxygeldanamycin) (tanespimycin)	P378/MEPK, EGFR pathway	Growth inhibition
Lung cancer	AUY922	AR and PI3K/mTOR RAF/MEK/ERK pathway	Antitumor activity
	CS-6	Targeting IKKβ/NF-κB pathway	Growth inhibition
Rheumatoid arthiritis, Inflammatory bowel disease, Osteoarthritis	Celastrol	RAF/MEK/ERK and PI3K/AKT/mTOR signaling pathways	Anti-inflammatory effect, Induce apoptosis
Prostate cancer, colon, and ovarian cancer	Gedunin	Disruptor of Hsp 90-p23 interaction	Ant proliferative activity
Gynecological cancer, Gastrointestinal cancer, Thyroid cancers and other cancers	Withaferin A	Hsp90-Cdc 37	Antitumor activity
	Derrubone	Disruptor of Hsp 90-Cdc 37 interaction	Anticancerous activity
	Cruentaren A	Hsp90/F1F0 ATP synthase disruptor	Antitumor activity. Highly cytotoxic to different cell lines

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
