# Peer review of "Hsp90 in Human Diseases: Molecular Mechanisms to Therapeutic Approaches"

_cells, 2022, doi:10.3390/cells11060976_

Round 1

Reviewer 1 Report

The writing of this article makes the reading easy and fluid. It's easy to follow the train of thought that leads from one section to the next. The work is well organized and ideas are presented clearly and concisely.

It is an original review that contributes to the construction of knowledge about the mechanisms by which treatments with Hsp90 inhibitors can generate the reported undesirable effects.

My recommendation is in favor of its publication. However, I have a minor criticism:

1) In line 249 there is an additional "and" just before the reference 71

2) In line 266 it is not clear to me what they mean by "AR and its transcription factors". AR is a transcription factor itself, are the authors referring to AR cofactors or associated chaperones?

3) At line 270, the authors refer to "AR apoptosis". It's a new term for me and I couldn't find it in the literature search I did. Are you referring to androgen receptor-mediated apostosis?

Author Response

We appreciate your comments. Please find our responses on the attached word file.

Reviewer 2 Report

The review written by Mamta P Sumi and Arnab Ghosh is very well organized and covers most of the diseases, where the small molecule HSP90 is involved. There are some suggestions that can help to improve the manuscript and make it more valuable. 

  1. HSPs are known to play an important role in the life cycle of protozoan parasites and nematode invasion, oogenesis, and replication. Some HSP90 inhibitors have shown opposing effects on Leishmania, Toxoplasma, and Theileria. These groups of diseases, especially zoonotic ones, should be also described in the manuscript. 
  2. There are a bunch of studies that demonstrated that second-generation HSP90 inhibitors, such as AUY-922 ameliorate pulmonary fibrosis in mouse models. Those studies should be mentioned in the review.

Author Response

We appreciate your comments for the betterment of the review. Please see our responses in the attached file.
